# Accelerating Sparse Autoencoder Training via Layer-Wise Transfer Learning in Large Language Models

**Davide Ghilardi[1] [*], Federico Belotti[1] [*], Marco Molinari[2, 4] [*], Jaehyuk Lim[2, 3]**

[1]University of Milan-Bicocca, [2]LSE.AI, [3]University of Pennsylvania, [4]London School of Economics

[*] Equal contribution

**Correspondence:** d.ghilardi@campus.unimib.it

## Abstract

Sparse AutoEncoders (SAEs) have gained popularity as a tool for enhancing the interpretability of Large Language Models (LLMs). However, training SAEs can be computationally intensive, especially as model complexity grows. In this study, the potential of transfer learning to accelerate SAEs training is explored by capitalizing on the shared representations found across adjacent layers of LLMs. Our experimental results demonstrate that fine-tuning SAEs using pre-trained models from nearby layers not only maintains but often improves the quality of learned representations, while significantly accelerating convergence. These findings indicate that the strategic reuse of pre-trained SAEs is a promising approach, particularly in settings where computational resources are constrained.

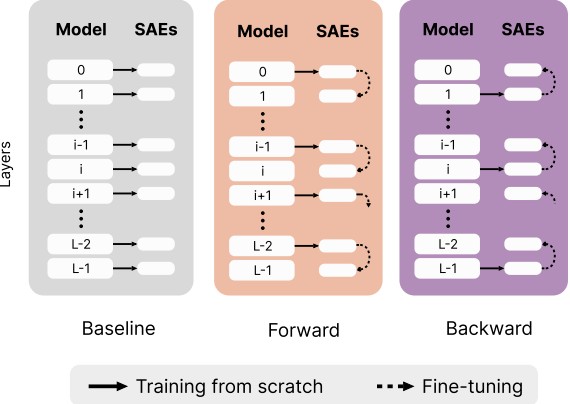

Figure 1: Visualization of our method. From left to right: **baseline** method where each Sparse AutoEncoder (SAE) is trained from scratch (solid line); **forward** method where SAEs are initialized with weights from the previous layer's SAE and fine-tuned (dashed line) with the new layer activations; **backward** method where SAEs are initialized with weights from the following layer's SAE.

## 1 Introduction

Transformer-based models have become ubiquitous in a large variety of different application fields (Dubey et al., 2024; Kirillov et al., 2023; Radford et al., 2023; Chen et al., 2021; Zitkovich et al., 2023; Waisberg et al., 2023). Given their tremendous impact on society, concerns about their interpretability have been raised by various stakeholders (Bernardo, 2023). Mechanistic Interpretability (MI) (Conmy et al., 2023; Nanda et al., 2023), seeks to reverse-engineer how Neural Networks, and in particular LLMs, generate outputs by uncovering the circuits they have learned during training, stored inside their parameters, and executed during a forward pass (Nanda et al., 2023; Conmy et al., 2023; Gurnee et al., 2023). A promising interpretability technique is dictionary learning (Cunningham et al., 2023; Gao et al., 2024; Karvonen et al., 2024) which seeks to capture interpretable and editable features within the internal layers of LLMs. This method implies training Sparse Autoencoders (SAEs) to reconstruct the model's activations using sparse learned features. However, training SAEs is computationally intensive, particularly when applied across multiple layers in deep networks. This computational burden poses a significant barrier to their widespread application, especially in resource-constrained environments where the cost of training from scratch is prohibitive. Recent research has highlighted the potential of transfer learning as a strategy to mitigate these challenges (Kissane et al., 2024). In particular, it has been shown in Gromov et al. (2024) that adjacent layers in LLMs are often redundant, suggesting that the knowledge encoded in one layer is also present in neighboring ones and that it can effectively be transferred. This observation forms the basis of our investigation: we hypothesize that SAEs trained on one set of layers can serve as effective initialization for SAEs designed for closely related layers. Specifically, the *forward* approach is defined as initializing an SAE with the weights of

a previous layer SAE, and the *backward* approach as initializing an SAE with the weights of a subsequent layer SAE. The overall training procedure is summarized in Figure 1. We tested this hypothesis on Pythia-160M, a small 12-layer decoder-only transformer from the Pythia family (Biderman et al., 2023). By reusing the representations learned in earlier layers, computational demands of training can be reduced by at least 25%[1] while maintaining, or even improving, the quality of the resulting models. Our contributions are as follows:

- We demonstrate that SAEs exhibit partial transfer to adjacent layers in a zero-shot setting, though fine-tuning is recommended for optimal performance.

- We show that both Forward-SAEs and Backward-SAEs, when fine-tuned on adjacent activations, consistently transfer across all tested checkpoints, achieving comparable or superior performance to SAEs trained from scratch, while using significantly less training data.

- We train and publicly release SAEs for Pythia-160M (Biderman et al., 2023), the model utilized in this study.

Code, data, and trained models will be publicly released after the double-blind review.

## 2  Background and objectives

### 2.1  Linear representation hypothesis and superposition

Although it has been demonstrated that LLMs represent some of their feature linearly (Park et al., 2024), a key challenge in LLM interpretability is the lack of clear neuron interpretation. Recent work of Elhage et al. (2022) tries to explain this phenomenon by showing that models can use $n$-dimensional activations to represent $m \gg n$ sparse almost-orthogonal features in *superposition*. Superposition theory is based on three key concepts: (i) the existence of a hypothetical large and disentangled model where each neuron perfectly aligns with a single feature, with each neuron activating for exactly one feature at a time. The observed models can be thought as dense, almost-orthogonal projections of this larger, ideal model. (ii) Features are

sparse, reflecting the idea that in the natural world, many features are inherently sparse. (iii) The importance of features varies depending on the task at hand. These assumptions, combined with two mathematical principles[2], suggest that the hidden sparse features can be recovered by projecting the dense model back to the hypothetical large and disentangled one. SAEs serve this purpose: learning a set of sparse, interpretable, and high-dimensional features from an observed model's dense and superposed activations.

### 2.2  Sparse Autoencoders

Recently, Sparse AutoEncoders have become a popular tool in Large Language Model (LLM) interpretability as they effectively decompose neuron activations into interpretable features (Bricken et al., 2023; Cunningham et al., 2023). For a given input activation $\mathbf{x} \in \mathbb{R}^{d_{\text{model}}}$, the SAE computes a reconstruction $\hat{\mathbf{x}}$ as a sparse linear combination of $d_{\text{sae}} \gg d_{\text{model}}$ features $\mathbf{v}_i \in \mathbb{R}^{d_{\text{model}}}$. The reconstruction is given by:

$$(\hat{\mathbf{x}} \circ \mathbf{f})(\mathbf{x}) = \mathbf{W}_d \mathbf{f}(\mathbf{x}) + \mathbf{b}_d \tag{1}$$

where $\mathbf{v}_i$ are the columns of $\mathbf{W}_d$, $\mathbf{b}_d$ is the bias term of the decoder and $\mathbf{f}(\mathbf{x})$ are feature activations. The latter are computed as:

$$\mathbf{f}(\mathbf{x}) = \text{ReLU}(\mathbf{W}_e(\mathbf{x} - \mathbf{b}_d) + \mathbf{b}_e) \tag{2}$$

where $\mathbf{b}_e$ is the encoder bias term. SAEs are trained to minimize the following loss function:

$$\mathcal{L}_{\text{sae}} = \|\mathbf{x} - \hat{\mathbf{x}}\|_2^2 + \lambda \|\mathbf{f}(\mathbf{x})\|_1 \tag{3}$$

In Equation 3, the first term corresponds to the reconstruction error, to which an $\ell_1$ regularization term on the activations $\mathbf{f}(\mathbf{x})$ is added to promote sparsity in the feature activations. In SAEs training, it is common to set $d_{\text{sae}} = c \, d_{\text{model}}$ with $c \in \{2^n \mid n \in \mathbb{N}_+\}$. So, the training process of a SAE can become computationally intensive, particularly as model size increases. For example, training a single SAE of a widely used model such as Llama-3-8b (Dubey et al., 2024) ($d_{\text{model}} = 4096$) with an expansion factor of $c = 32$ (i.e., $d_{\text{sae}} = 131072$) requires $\approx$ 1B parameters. Under these circum-

---

[1]Assuming training half of SAEs from scratch and the other half with transfer from an adjacent layer with half of the training tokens.

[2]The Johnson-Lindenstrauss lemma, which ensures that points in a high-dimensional space can be embedded into a lower dimension while almost preserving distances, and compressed sensing, which exploits sparsity to recover signals from fewer samples than required by the Nyquist–Shannon theorem

| Config | Value |
|---|---|
| Layers ($L$) | 12 |
| Model dimension ($d_{\text{model}}$) | 768 |
| Heads ($H$) | 12 |
| Non-Embedding params | 85,056,000 |
| Equivalent models[3] | GPT-Neo OPT-125M |

Table 1: Pythia-160M model specifics

stances, transfer learning is a useful resource to reduce the number of trained SAEs, with the transfer that can happen *intra-model*, where SAEs training is shared between layers of the same model (our case), or *inter-model*, where SAEs are shared between different fine-tuned versions of the same model as shown in Kissane et al. (2024).

## 2.3 Evaluating SAEs

Evaluating SAEs and the features they have learned presents significant challenges. In our work, the techniques employed can be divided into *reconstruction* and *interpretability* metrics. The first includes:

- The Cross-Entropy Loss Score (CES), is defined as

$$\text{CES} = \frac{\text{CE}(\zeta) - \text{CE}(\hat{\mathbf{x}} \circ \mathbf{f})}{\text{CE}(\zeta) - \text{CE}(\text{Id})} \quad (4)$$

  where $\hat{\mathbf{x}} \circ \mathbf{f}$ is the autoencoder function, $\zeta : \mathbf{x} \to \mathbf{0}$ the zero-ablation function and $\text{Id} : \mathbf{x} \to \mathbf{x}$ the identity function. According to this definition, a SAE would get a CES equal to 1 if it perfectly reconstructs $\mathbf{x}$ ($> 1$ if it improves the CE loss), $\leq 0$ when the reconstruction is not better than zero-ablation, otherwise the score is comprised in the unit interval.

- The $L2$ loss (reconstruction loss) is the first term of Equation 3, which measures the reconstruction error made by the SAE.

- The $L0$ loss of the learned features, defined as

$$\|\mathbf{f}\|_0 = \sum_{j=1}^{|\mathbf{f}|} \mathbb{I}[\mathbf{f}_j \neq 0] \quad (5)$$

which represents the number of non-zero SAE features used to compute the reconstruction.

Measuring the quality of the features learned by a SAE is not straightforward, and multiple strategies exist. As reported in Makelov et al. (2024), *interpretability* metrics can be categorized as follows:

- Indirect Geometric Measures: Sharkey et al. (2023) proposed using mean maximum cosine similarity (MMCS) between features learned by different SAEs to assess their quality. Given two feature dictionaries $D$ and $D'$, with $|D| = |D'|$, MMCS is defined as:

$$\text{MMCS}_{D,D'} = \frac{1}{|D|} \sum_{\mathbf{u} \in D} \max_{\mathbf{v} \in D'} \text{CosSim}(\mathbf{u}, \mathbf{v}) \quad (6)$$

- Auto-Interpretability: Bricken et al. (2023), Bills et al. (2023), and Cunningham et al. (2023) used LLMs to generate natural-language descriptions of SAE features based on highly activating examples and measured interpretability as the prediction quality on previously unseen text.

- Manually Crafted Proxies for Ground Truth: (Bricken et al., 2023) developed computational proxies for a set of SAE features, relying on manually formulated hypotheses.

- Faithfulness and Completeness of task feature circuits: Marks et al. (2024) compute faithfulness and completeness as measures to estimate the task sufficiency and necessity of learned SAE features. In particular, given a task, they first compute a circuit $C$ of SAE features by selecting them according to their importance, estimated via their Indirect Effect[4] (Pearl, 2022):

$$\begin{aligned} \text{IE}(m; \mathbf{f}; a_c, a_w) = \\ m[M(a_c|\text{do}(\mathbf{f} = \mathbf{f}_w), x); M(a_c|x)] \quad (7) \end{aligned}$$

where $x$ is a given prompt and $m : \mathbb{R}^{d_{\text{vocab}}} \to \mathbb{R}$ is the logit-difference computed by a LLM $M$ over two contrastive answer tokens $a_c, a_w$.[5] In this equation, $\mathbf{f}_w$ represents SAE feature activations during the computation of

---

[3]As specified in (Biderman et al., 2023)

[4]We estimate the IE through Attribution Patching (AtP) (Syed et al., 2023; Nanda, 2023) A formal definition of AtP is given in Appendix A

[5]E.g., $x =$ "The square root of 9 is", $a_c = 3$, and $a_w = 2$

$M(a_w|x)$, and $M(a_c|\mathrm{do}(\mathbf{f} = \mathbf{f}_w), x)$ refers to the value of $M(a_c)$ under an intervention where the activation of feature $\mathbf{f}$ is set to $\mathbf{f}_w$. Then, they estimate the *faithfulness* as

$$\frac{m(C) - m(\emptyset)}{m(M) - m(\emptyset)} \qquad (8)$$

where $m(C)$ is the model logit difference when using only the important SAE features while mean-ablating the others; $m(M)$, $m(\emptyset)$ represent the logit-difference achieved by the model alone and with the mean-ablated SAE reconstructions, respectively. *Completeness* is estimated by replacing $m(C)$ with $m(M \setminus C)$ in Equation 8. Intuitively, faithfulness captures the proportion of the model's performance the circuit $C$ explains, relative to mean-ablate the full model, thus modeling sufficiency. On the other hand, completeness captures the necessity of the learned features by measuring low downstream performance whenever the important SAE features are mean-ablated.

- Supervised Dictionary Benchmarking: Makelov et al. (2024) introduced a technique that benchmarks unsupervised SAE dictionaries against supervised dictionaries based on task-relevant attributes to ensure extracted features are interpretable and relevant to specific tasks.

In our work, evaluation metrics employed include all the reconstruction techniques listed above, the MMCS between features from SAEs trained with transfer learning and the ones from SAEs trained from scratch, and a Human Interpretability Score defined in Section 3. Moreover, we evaluate both faithfulness and completeness on three standard downstream tasks: Indirect Object Identification (IOI) (Wang et al., 2023), Greater Than (Hanna et al., 2023), and Subject-Verb Agreement (Marks et al., 2024), all of them comprising a set of examples in the form of $\{(x, a_c, a_w)_i\}$. Additionally, for faithfulness and completeness computation we fix the number of top important features $N$ throughout all the experiments: for faithfulness we let $N$ vary in $\{123, 246, 368, 492\}$, which correspond to $2\%, 4\%, 6\%$ and $8\%$ of top active features; for completeness, $N$ varies in $\{4, 36, 68, 100\}$.[6] Finally, in Appendix B we report the Direct Logit

---

[6]Top important features are computed on a per-example basis.

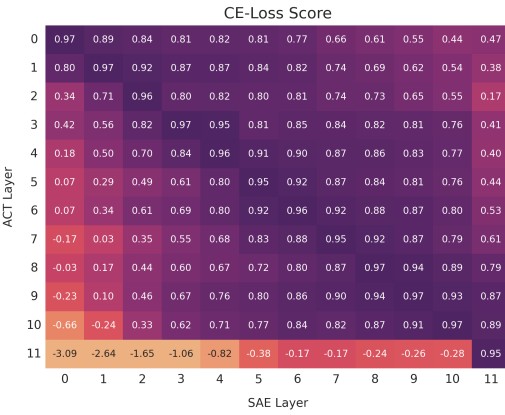

Figure 2: Cross-Entropy Loss Score (CE-Loss Score) (Eq. 4), where the cell $(i, j)$ in the plot represents the CE-Loss Score obtained by reconstructing the activations from layer $i$ with $\mathrm{SAE}_j$. This plot has to be read column-wise.

Attribution (DLA), as specified by Bricken et al. (2023).

## 2.4 Transfer Learning

Transfer learning (Goodfellow et al., 2016) is a powerful technique in machine learning where knowledge gained from one task is applied to improve performance on a related, but distinct, task. This approach is particularly useful when training from scratch is computationally expensive or when labeled data is scarce. In the context of SAEs for LLMs, transfer learning enables the reuse of weights learned in one layer to initialize and accelerate the training of SAEs in adjacent layers.

## 2.5 Objectives

In this work transferability and generalization of intra-model SAEs have been studied, aiming to answer the following research questions:

**Q1**. Are SAEs transferable between layers? I.e., can a SAE trained on the activations of layer $i$ be reused to reconstruct activations of layer $j \neq i$?

**Q2**. Is Transfer Learning applicable to SAEs? Specifically, can a SAE initialized with the weights of a neighboring SAE and then fine-tuned achieve equal or superior performance, potentially using only a fraction of the data, compared to an SAE trained from scratch?

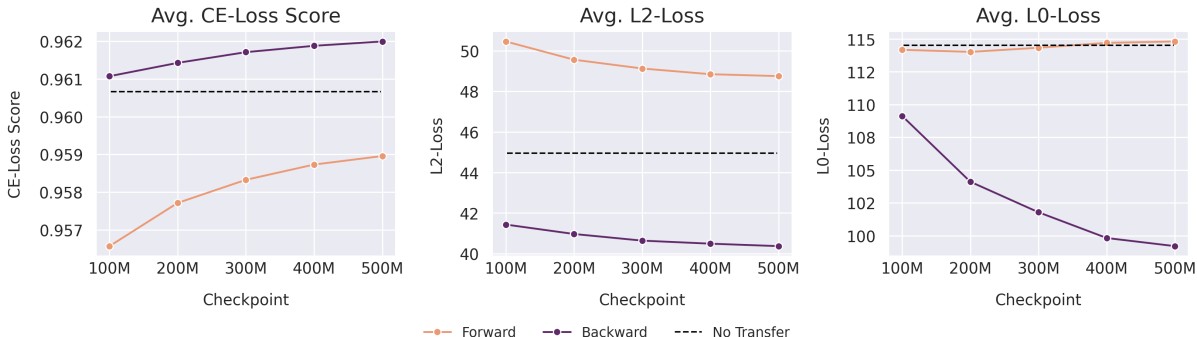

Figure 3: Average CE-Loss Score, $L2$-Loss and $L0$-Loss. The average is computed over layers for a single checkpoint. The "No Transfer" average is computed considering the performance obtained by $\text{SAE}_i(\mathbf{x}_i), \forall i = 0, ..., 11$.

## 3 Experimental setup

To address the questions raised in Section 2, we first trained from scratch one $\text{SAE}_i$ for each layer $i$ of Pythia-160M, a 12-layer decoder-only Transformer model from the Pythia family (Biderman et al., 2023). Each SAE was trained using the JumpReLU activation function (Rajamanoharan et al., 2024), with activations taken from the corresponding layer's residual stream after the MLP contribution. The model configuration details are provided in Table 1. Let also $j \neq i$ be another layer index. Then $\text{SAE}_{i \leftarrow j}$ is defined as the SAE initialized with weights from the $j$-th SAE and fine-tuned with activations of the $i$-th layer. In particular, this work is focused on $\text{SAE}_{i \leftarrow i-1}$ and $\text{SAE}_{i \leftarrow i+1}$, named Forward-SAE (Fwd-SAE) and Backward-SAE (Bwd-SAE) respectively. Figure 1 summarizes the overall training and fine-tuning procedure, with the hyperparameters specified in Table 2. The dataset adopted for both training and fine-tuning is the Pile-small-2b[7], an already tokenized version of the Pile dataset (Gao et al., 2020) with a total of 2b tokens. To effectively measure the reconstruction performance of a SAE before and after fine-tuning with transfer learning, the normalized CE-Loss Score is adopted and defined as:

$$\overline{\text{CES}_{i,j}} = \frac{\text{CES}(\text{SAE}_{i \leftarrow j}(\mathbf{x}_i)) - \text{CES}(\text{SAE}_j(\mathbf{x}_i))}{\text{CES}(\text{SAE}_i(\mathbf{x}_i)) - \text{CES}(\text{SAE}_j(\mathbf{x}_i))} \quad (9)$$

by assuming $\text{CES}(\text{SAE}_j(\mathbf{x}_i))$ and $\text{CES}(\text{SAE}_i(\mathbf{x}_i))$ being, respectively, the lower and the upper bound for the CES on $\mathbf{x}_i$. With the definitions above, $\overline{\text{CES}_{i,i-1}}$ and $\overline{\text{CES}_{i,i+1}}$ are the normalized CE-Loss Score of the Fwd-SAE and Bwd-SAE re-

[7] https://huggingface.co/datasets/NeelNanda/pile-small-tokenized-2b

spectively. Finally, to evaluate feature quality, a *Human Interpretability Score* has been defined as the ratio of features that have been evaluated interpretable by human annotators. To generate the score, all the SAEs have been run on approximately 1M tokens randomly sampled from the training dataset. With their activations, max activating tokens and top/bottom attribution logits have been computed and analyzed from the labelers.

## 4 Results

### 4.1 SAE transferability

Figure 2 shows the CE-Loss Score achieved by every $\text{SAE}_j$ reconstructing the activations of layer $i$, for every $i, j = 0, ..., L - 1$, i.e., the zero-shot setting. It is clear that a certain degree of transferability exists between $\text{SAE}_j$ and the activations of adjacent layers, with this being more noticeable when $i = j - 1$ (i.e., SAEs are more effective at reconstructing the activations of preceding layers than those of subsequent ones). These findings can also be attributed to the fact that, as demonstrated by Gromov et al. (2024), angular distances between adjacent layers are smaller, enabling neighboring SAEs to operate on a similar basis with respect to the activations they were trained on. The answer to **Q1** is, therefore, yes; however, although transferability between layers exists, it remains partial and, potentially, not completely reliable for downstream applications.

### 4.2 SAE transfer learning

Figure 3 shows all reconstruction metrics averaged for all layers across every tested checkpoint. Detailed results for single layer and aggregated over time can be found in Appendix C (Figures 9 - 17) along with the normalized CE-Loss Score

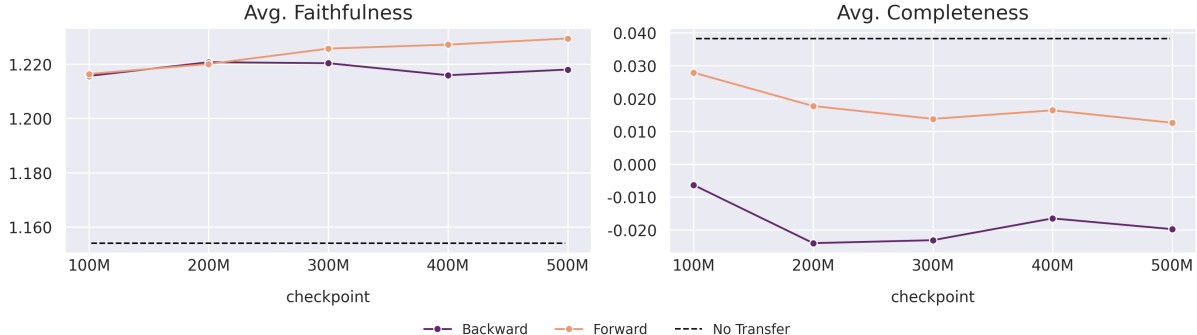

Figure 4: Average Faithfulness and Completeness. The average is computed over layers and the number of important active SAE features for a single checkpoint. The "No Transfer" average is computed considering the performance obtained by $\text{SAE}_i(\mathbf{x}_i), \forall i = 0, ..., 11$.

(Eq. 9) in Tables 3 and 4. Looking at the plots, it can be seen that forward and backward SAEs achieve almost equal or even superior performance than the ones trained from scratch with as little as $1/10$-th (100M tokens) of the original training data (1B tokens), with the scores constantly increasing with the number of tokens used for fine-tuning. As a result, it can be said that both forward ad backward are effective strategies to reduce the number of SAEs trained from scratch. Between the two, the backward technique is the one that constantly shows better results, both in terms of CE-Loss Score, $L2$, and $L1$ loss. So, the answer to **Q2** is also yes if we just consider the reconstruction metrics. To fully respond to **Q2** beyond reconstruction performance, the quality of the learned SAE features have to be inspected.

### 4.3 Feature Evaluation

Figure 4 displays the layer-averaged faithfulness and completeness scores for each tested checkpoint. The plot reveals that both forward and backward

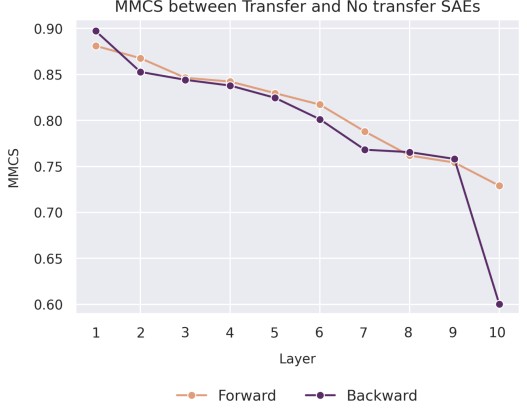

Figure 5: Per-layer MMCS of the Forward and Backward SAEs.

transfer SAEs consistently achieve better scores than the baseline SAEs, with minimal differences between the two transfer methods. Therefore, both the forward and backward SAEs maintain sufficiency and necessity during their transfer. Figure 5 presents the MMCS between SAEs trained with transfer learning and those trained from scratch. The metric value decreases for deeper layers, suggesting a slight divergence in the features learned by the transfer SAEs. Notably, $\text{SAE}_{L-1 \leftarrow L}$ exhibits a sharp decline in the score, indicating that transferring on the last layer should be approached with caution. Lastly, from human interpretability scores (Figure 7), no significant differences can be observed between each transfer type. By manually looking at the learned features, a key pattern has emerged: many features learned by SAEs trained with transfer learning remain shared with the SAE used for initialization. This phenomenon, termed *Feature Transfer*, particularly affects the most interpretable features (see an example in Figure 23). To further investigate this phenomenon, a metric was developed to quantify it. Given a $\text{SAE}_i$ and another trained via transfer learning from it, $\text{SAE}_{i \leftarrow i \pm 1}$, the number of shared "top", "bottom", and "max activating tokens"[8] for each feature have been computed (features have been compared using the same indices). The transfer score has been then defined as the percentage of shared tokens across all three heuristics. Figure 6 presents the scores across all the layers for the last evaluated checkpoint. Except for layer 1, backward transfer consistently exhibits lower scores. It's important to note that this phe-

---

[8]"Top" and "bottom" logit tokens refer to those whose unembedding directions are most and least aligned, respectively, with the projection of the feature in the unembedding space. "Max activating" tokens are those for which the feature exhibits the highest activations.

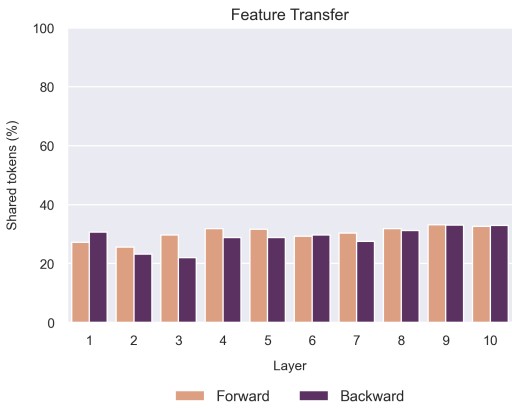

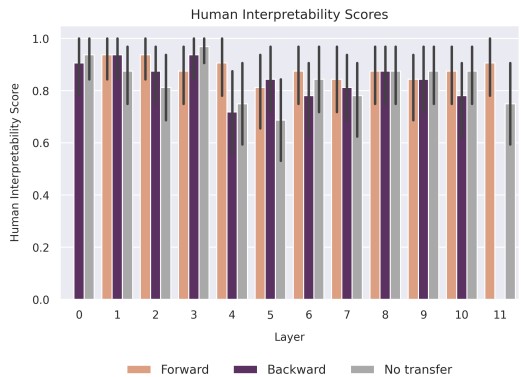

Figure 6: Per-layer number of shared tokens for the Forward and Backward SAEs, as defined in Section 4.3. Each bar represents the percentage of shared token between $SAE_i$ trained from scratch and forward $SAE_{i+1\leftarrow i}$ and backward $SAE_{i-1\leftarrow i}$, respectively.

Figure 7: Human Interpretability Scores (Section 3) for 32 features randomly sampled from each SAE layer and type of transfer.

nomenon is easily recognized in SAEs trained with transfer learning when compared to their initialization, as feature indices are preserved. Evaluating this in SAEs trained from scratch is more demanding due to the exponential growth in the number of comparisons required, and although relevant, it falls outside the scope of this work.

### 4.4 Compute Efficiency

Leveraging forward and backward transfer, we were able to reduce total training steps when utilizing forward transfer and backward transfer by 42% and 46%, respectively. Check Appendix B.1 for details.

## 5 Related works

### 5.1 Scaling and evaluating SAEs

As SAEs gain popularity for LLMs interpretability and are increasingly applied to state-of-the-art models (Lieberum et al., 2024), the need for more efficient training techniques has become evident. To address this, Gao et al. (2024) explored scaling laws of autoencoders to identify the optimal combination of size and sparsity. However, training SAEs is only one aspect of the challenge; evaluating them presents another significant hurdle. This evaluation is a crucial focus within MI. While early approaches in Cunningham et al. (2023) and (Bricken et al., 2023) relied on unsupervised metrics like reconstruction loss and $L0$ sparsity to assess SAE performance, these metrics alone cannot fully capture the efficacy of a SAE. They provide quantitative measures of how well SAEs capture informa-

tion in model activations while maintaining sparsity, but they fall short of addressing the broader utility of these features. More recent techniques, such as auto-interpretability (Bricken et al. (2023), Bills et al. (2023), Cunningham et al. (2023)) and ground-truth comparisons (Sharkey et al., 2023), have shifted towards a more holistic evaluation, focusing on the causal relevance of the extracted features (Marks et al., 2024) and evaluating SAEs on different downstream tasks in which they can be employed (Makelov et al., 2024). In particular, Makelov et al. (2024) introduced a framework for evaluating SAEs on the Indirect Object Identification (IOI) task, focusing on three key aspects: the sufficiency and necessity of activation reconstructions, the ability to control model behavior through sparse feature editing, also called feature steering (Templeton et al., 2024), and the interpretability of features in relation to their causal role. Karvonen et al. (2024) further advanced principled evaluations by introducing novel metrics specifically designed for board game language models. Their approach leverages the well-defined structure of chess and Othello to create supervised metrics for SAE quality, including board reconstruction accuracy and coverage of predefined board state properties. These methods provide a more direct assessment of how well SAEs capture semantically meaningful and causally relevant features, offering a complement to the earlier unsupervised metrics like $L0$ and $L2$.

### 5.2 SAEs transfer learning

Recent work by Kissane et al. (2024) and Lieberum et al. (2024) has demonstrated the transferability of SAE weights between base and instruction-tuned

versions of the Gemma-1 (Team et al., 2024a) and Gemma-2 (Team et al., 2024b), respectively. This finding is significant as it suggests that many interpretable features are preserved during the fine-tuning process. While this transfer occurs between model variants (inter-model) rather than between layers (intra-model), it complements our work by indicating that SAE features can remain stable across different stages of model development. The preservation of these features through fine-tuning not only offers insights into the robustness of learned representations but also suggests potential efficiency gains in interpreting families of models derived from a common base SAE.

## 6 Conclusions

We hypothesized and validated whether SAE transfer is an effective method to accelerate and optimize the SAE training process. We investigated whether SAE weights derived from adjacent layers could maintain efficacy in reconstruction, which our results affirmed. Furthermore, we examined whether the transferred SAEs, when fine-tuned on a layer's activations, could reliably capture monosemantic features comparable to the original SAE, which has been also confirmed by our experiments. The transferred SAEs (both forward and backward) demonstrated comparable and occasionally superior reconstruction loss relative to the original. Empirically, we observed frequent overlap in the most strongly activated features across adjacent layers (e.g. Figure 23). For a given feature index $i$, the features learned by $\text{SAE}_{i \leftarrow i+1}$ (Backward), $\text{SAE}_i$ (No Transfer), and $\text{SAE}_{i \leftarrow i-1}$ (Forward) appeared to represent similar concepts.

## 7 Limitations and future works

While our study successfully demonstrates the feasibility of reconstruction transfer and the transfer learning of SAE weights to adjacent layers, there are several limitations that warrant consideration and pave the way for future research directions.

- *Model Size and Scope*: We trained base and transfer SAEs on the activations of Pythia-160m, a model mcuh smaller than state-of-the-art LLMs. Although not being tested, as model size and training complexity increase, the benefits of transfer learning are expected to become more pronounced. In such scenarios, transfer learning can significantly accelerate training and reduce associated costs,

making our approach potentially more impactful for larger models. Therefore, a critical area for future research is to extend these investigations to larger models, exploring how scaling affects the efficacy of transfer learning and how these benefits can be maximized in real-world settings.

- *Inter-Model and Intra-Model transferability*: In our study, we focused on the transfer of intra-model SAEs, particularly assessing the transferability between SAEs in adjacent layers. Given that model architectures are now commonly shared across different model families, a direction for future research would be to evaluate the transferability of intra-model SAEs within models from different families that utilize the same architecture. This exploration could offer valuable insights into the broader applicability of SAEs beyond closely related model families.

- *Experimental Scale and Hyperparameter Interactions*: Our study was conducted on a limited scale in terms of model components involved and the range of training hyperparameters explored. The fixed set of hyperparameters used may not fully capture the potential of our transfer learning approach across different configurations. Future research should involve a broader exploration of hyperparameter spaces, especially the $\lambda$ coefficient and expansion factor $c$, along with component variations to determine the robustness and versatility of the method.

- *Feature Transfer Phenomenon*: Our findings reveal a "feature transfer" phenomenon, where features learned in one layer are exactly replicated in another during transfer learning. This can be problematic, as it may prevent the fine-tuned SAEs from discovering new, layer-specific features. However, it also offers an interesting opportunity to study how similar features are encoded across layers. Future research should focus on understanding and managing this phenomenon to either harness or mitigate its effects, depending on the desired outcomes, thereby improving the flexibility and effectiveness of transfer learning.

## Acknoledgements

This work has been partially funded by the European innovation action enRichMyData (HE 101070284).

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

## A    IE estimation through Attribution Patching

In Equation 7 we reported the Indirect Effect (IE) (Pearl, 2022), which measures the importance of a feature with respect to a generic downstream task $\mathcal{T}$. To reduce the computational burden of estimating the IE with a single forward pass per feature, we employed Attribution Patching (AtP) (Nanda, 2023; Syed et al., 2023). AtP employs a first-order Taylor expansion

$$\hat{\text{IE}}_{\text{AtP}}(m; \mathbf{f}; a_c, a_w) = \nabla_{\mathbf{f}} m \big|_{\mathbf{f}=\mathbf{f}_c} (\mathbf{f}_w - \mathbf{f}_c) \quad (10)$$

which estimates Equation 7 for every $\mathbf{f}$ in two forward passes and a single backward pass.

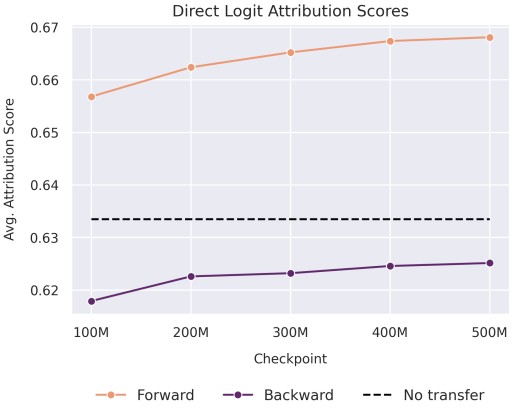

Figure 8: Direct Logit Attribution Scores averaged across layers for every tested checkpoint compared to the "No Transfer" baseline, i.e. the DLA scores obtained by $\text{SAE}_i(\mathbf{x}_i), \forall i = 0, ..., 11$.

## B    Direct Logit Attribution

We also report the Direct Logit Attribution (DLA) between forward $\text{SAE}_{i \leftarrow i-1}$ and backward $\text{SAE}_{i \leftarrow i+1}$ transfer SAEs. Introduced by Bricken et al. (2023), DLA assesses the direct effect of a feature on the next-token distribution, providing insights into the causal role of features. The attribution score is computed as follows:

$$\text{attr}_i(x; a_c; a_w) = \mathbf{f}_i \, \mathbf{v}_i \cdot \nabla_x \mathcal{L}(a_c, a_w) \quad (11)$$

where $x$ is a given prompt and $\nabla_x \mathcal{L}$ is the gradient of the logit difference between two contrastive answer tokens $a_c, a_w$ (E.g., $x =$ "The square root of 9 is", $a_c = 3$, and $a_w = 2$). We report the feature averaged DLA computed on a custom dataset comprising 64 handcrafted prompts in the form of $\{(x, a_c, a_w)_i\}$. Figure 8

displays the layer-averaged DLA scores for each tested checkpoint. The plot reveals that forward transfer SAEs consistently achieves higher scores than the baseline, while backward transfer SAEs consistently scores lower. This outcome contrasts with the reconstruction metrics, where the backward technique consistently outperformed the forward approach. A detailed per-layer DLA scores plot is reported in Figure 22.

### B.1    Compute Efficiency

This work proposes a novel method leveraging transfer learning to significantly reduce computational costs in training SAEs in the context of LLMs. We demonstrate that both Fwd-SAE $\text{SAE}_{i \leftarrow i-1}$ and Bwd-SAE $\text{SAE}_{i \leftarrow i+1}$, trained with our fine-tuning strategy, are both valid alternatives to the standard layer-by-layer training of $\text{SAE}_i$, in terms of both reconstruction quality of the learned representation and performance on downstream tasks. In practice, our approach consists of the following steps:

1. Train a $\text{SAE}_i$ on alternate layers, depending on the transfer direction. For Forward transfer $i \in \{0, 2, 4, ..., L\}$, while for Backward transfer $i \in \{1, 3, 5, ..., L-1\}$.

2. Initialize the current $\text{SAE}_i$ by either $\text{SAE}_{i \leftarrow i-1}$ for forward transfer or $\text{SAE}_{i \leftarrow i+1}$ for backward transfer.

3. Apply transfer learning by training the remaining SAEs and stop when some criteria are matched (e.g., when the loss converges to a specific value or when a computational budget has been reached).

Empirical results demonstrate substantial efficiency gains. In our experiments with a 12-layer Pythia-160M (Biderman et al., 2023) model, we observed a performance increase after fine-tuning on 10% of the training data (Figure 3 and Figure 4), with performance increasing over time. Extrapolating these findings, we can compute empirical lower and upper bounds on the training efficiency. Given a model with $L$ (in our particular case $L = 12$) layers and a training set consisting of 1B tokens, we have:

- **Baseline training:** Train one $\text{SAE}_i \ \forall i \in \{1, ..., 12\}$ for 1B tokens: 12B tokens

- **Forward/Backward transfer - 10% of data**:

- Train one $SAE_i$ for half of the layers for 1B tokens: 6B tokens
- Fine-tune the remaining $SAE_{i \leftarrow i-1}$ or $SAE_{i \leftarrow i+1}$ for 100M tokens: 0.6B tokens
- **Total**: 6.6B tokens

- **Forward/Backward transfer - 50% of data**:
  - Train one $SAE_i$ for half of the layers for 1B tokens: 6B tokens
  - Fine-tune the remaining $SAE_{i \leftarrow i-1}$ or $SAE_{i \leftarrow i+1}$ for 500M tokens: 3B tokens
  - **Total**: 9B tokens

- **Computational savings:**
  - **Lower bound** Forward/Backward transfer: 12B - 6.6B = 5.4B tokens
  - **Upper bound** Forward/Backward transfer: 12B - 9B = 3B tokens

- **Relative reduction in compute cost:**
  - **Lower bound** Forward/Backward transfer: $\frac{5.4B}{12B} \times 100\% = 45\%$
  - **Upper bound** Forward/Backward transfer: $\frac{9B}{12B} \times 100\% = 25\%$

Our analysis indicates that the proposed transfer learning approach can reduce compute costs by 25% to 45% for forward and backward transfer when fine-tuned for 50% and 10% of the training data respectively, improving efficiency and reducing costs by a great margin, while maintaining both reconstruction quality and performance on downstream tasks.

## C   Additional plots and tables

| Hyperparameter | Value |
| --- | --- |
| c | 8 |
| $\lambda$ | 1.0 |
| Hook name | resid-post |
| Batch size | 4096 |
| Adam $(\beta_1, \beta_2)$ | $(0, 0.999)$ |
| lr (Train) | 3e-5 |
| lr (Fine-tuning) | 1e-5 |
| lr scheduler | constant |
| lr deacy steps | 20% of the training steps |
| l1 warm-up steps | 5% of the training steps |
| # tokens (Train) | 1B |
| # tokens (Fine-tuning) | 500M |
| Checkpoint freq. | 100M |

Table 2: Training and fine-tuning hyperparameters

| Checkpoint | $i$ | | | | | | | | | | |
|---|---|---|---|---|---|---|---|---|---|---|---|
| | **1** | **2** | **3** | **4** | **5** | **6** | **7** | **8** | **9** | **10** | **11** |
| 100M | 0.962 | 0.960 | 0.983 | 0.920 | 0.865 | 0.439 | 0.955 | 0.948 | 0.858 | 0.944 | 1.003 |
| 200M | 0.968 | 0.968 | 0.996 | 0.933 | 0.873 | 0.459 | 0.970 | 0.956 | 0.894 | 0.965 | 1.005 |
| 300M | 0.969 | 0.971 | 1.000 | 0.941 | 0.877 | 0.475 | 0.981 | 0.960 | 0.911 | 0.972 | 1.005 |
| 400M | 0.971 | 0.974 | 1.003 | 0.944 | 0.879 | 0.479 | 0.988 | 0.963 | 0.921 | 0.978 | 1.006 |
| 500M | 0.972 | 0.975 | 1.005 | 0.946 | 0.881 | 0.488 | 0.991 | 0.964 | 0.929 | 0.981 | 1.006 |

Table 3: Normalized CE-Loss Scores $\overline{\mathrm{CES}_{i,i-1}}$ (Eq. 9) of the Fwd-SAE at different checkpoints. On $i = 6$, the Normalized CE-Loss Score increases over time even though it starts with a lower value w.r.t. the other checkpoints. From Figure 9 we note how the CE-Loss Score of $\mathrm{SAE}_5(\mathbf{x}_6)$ and $\mathrm{SAE}_{6\leftarrow 5}(\mathbf{x}_6)$ are nearly identical to the obtained by $\mathrm{SAE}_6(\mathbf{x}_6)$, thus the increment given by the fine-tuning over the baseline $\mathrm{SAE}_5(\mathbf{x}_6)$, captured by the Normalized CE-Loss Score in Eq. 9, is minimal and resulting in a lower value.

| Checkpoint | $i$ | | | | | | | | | | |
|---|---|---|---|---|---|---|---|---|---|---|---|
| | **0** | **1** | **2** | **3** | **4** | **5** | **6** | **7** | **8** | **9** | **10** |
| 100M | 0.988 | 0.927 | 0.964 | 1.052 | 0.803 | 0.375 | 0.801 | 1.044 | 0.920 | 1.005 | 0.939 |
| 200M | 0.990 | 0.939 | 0.969 | 1.076 | 0.812 | 0.396 | 0.805 | 1.047 | 0.912 | 1.001 | 0.953 |
| 300M | 0.991 | 0.945 | 0.972 | 1.084 | 0.823 | 0.412 | 0.808 | 1.049 | 0.913 | 0.999 | 0.965 |
| 400M | 0.995 | 0.951 | 0.975 | 1.098 | 0.827 | 0.420 | 0.811 | 1.052 | 0.912 | 0.997 | 0.972 |
| 500M | 0.997 | 0.951 | 0.975 | 1.098 | 0.827 | 0.425 | 0.814 | 1.056 | 0.913 | 0.998 | 0.976 |

Table 4: Normalized CE-Loss Scores $\overline{\mathrm{CES}_{i,i+1}}$ of the Bwd-SAE at different checkpoints. On $i = 5$, the Normalized CE-Loss Score increases over time even though it starts with a lower value w.r.t. the other checkpoints. From Figure 9 we note how the CE-Loss Score of $\mathrm{SAE}_6(\mathbf{x}_5)$ and $\mathrm{SAE}_{5\leftarrow 6}(\mathbf{x}_5)$ are nearly identical to the obtained by $\mathrm{SAE}_5(\mathbf{x}_5)$, thus the increment given by the fine-tuning over the baseline $\mathrm{SAE}_6(\mathbf{x}_5)$, captured by the Normalized CE-Loss Score in Eq. 9, is minimal and resulting in a lower value.

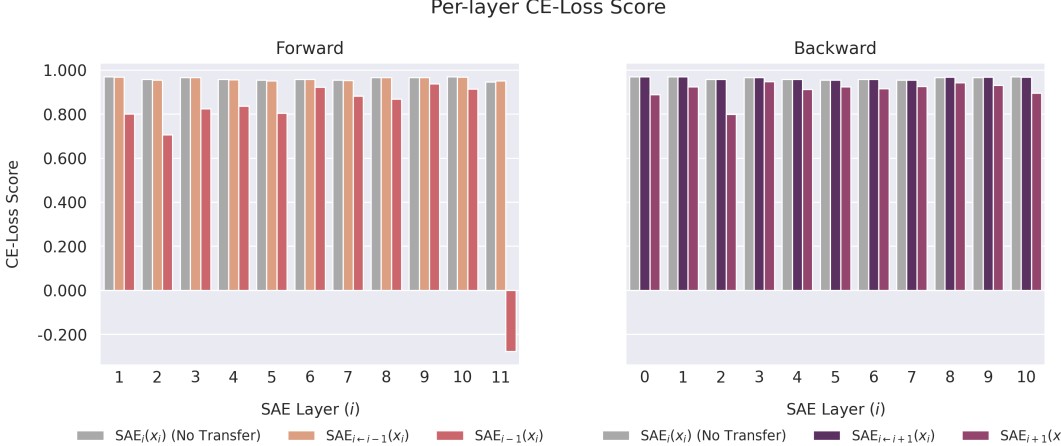

Figure 9: Detailed per-layer CE-Loss Score at the final checkpoint (500M). $\text{SAE}_{i-1}(\mathbf{x}_i)$ and $\text{SAE}_{i+1}(\mathbf{x}_i)$ are the baselines for the Fwd-SAE and Bwd-SAE respectively.

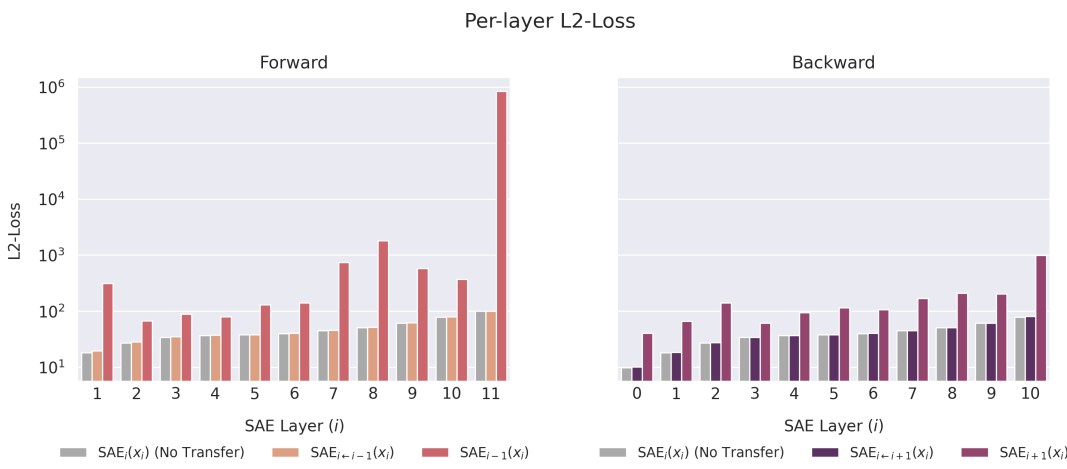

Figure 10: Detailed per-layer $L2$-Loss at the final checkpoint (500M). $\text{SAE}_{i-1}(\mathbf{x}_i)$ and $\text{SAE}_{i+1}(\mathbf{x}_i)$ are the baselines for the Fwd-SAE and Bwd-SAE respectively. The $y$-axis is on a logarithmic scale.

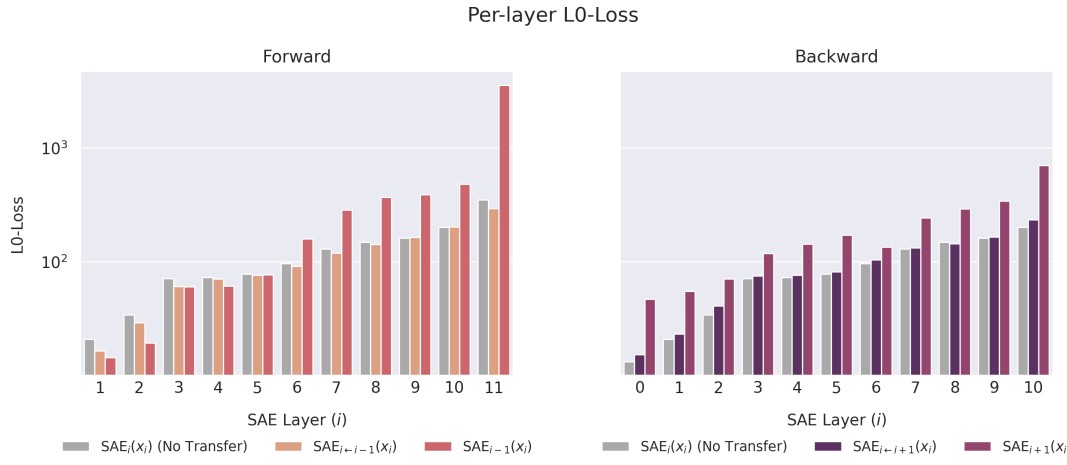

Figure 11: Detailed per-layer $L0$-Loss at the final checkpoint (500M). $\text{SAE}_{i-1}(\mathbf{x}_i)$ and $\text{SAE}_{i+1}(\mathbf{x}_i)$ are the baselines for the Fwd-SAE and Bwd-SAE respectively. The $y$-axis is on a logarithmic scale.

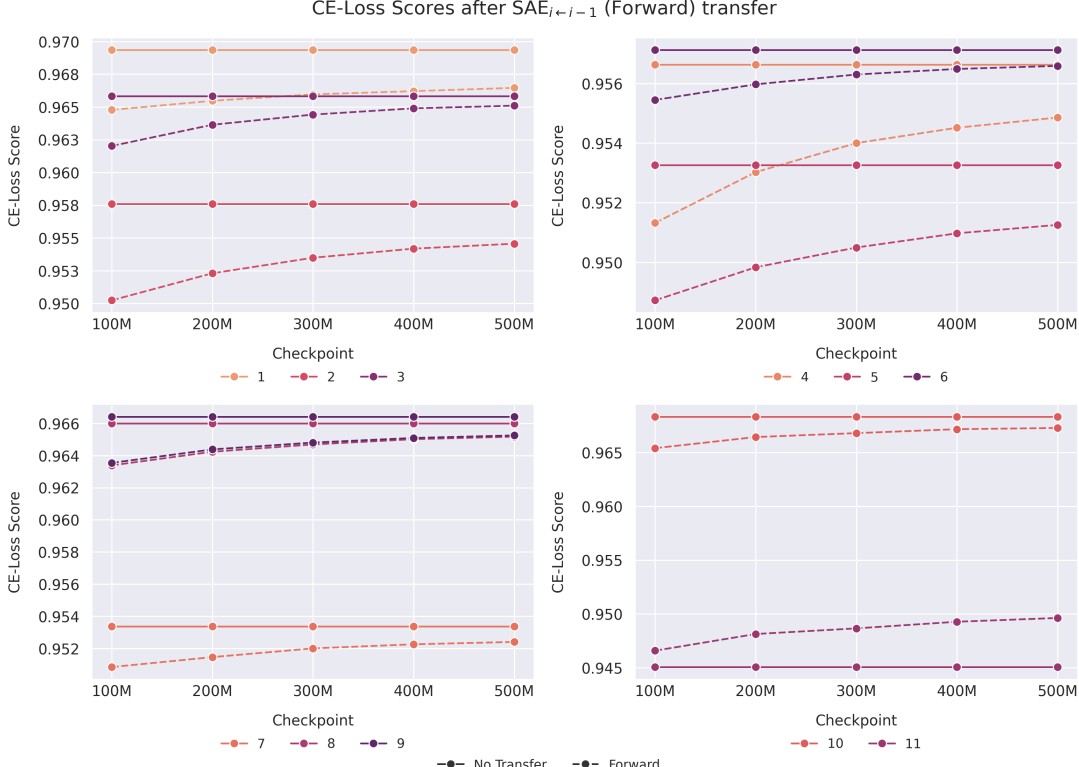

Figure 12: Detailed per-layer CE-Loss Score over time (Checkpoint) after Forward Transfer.

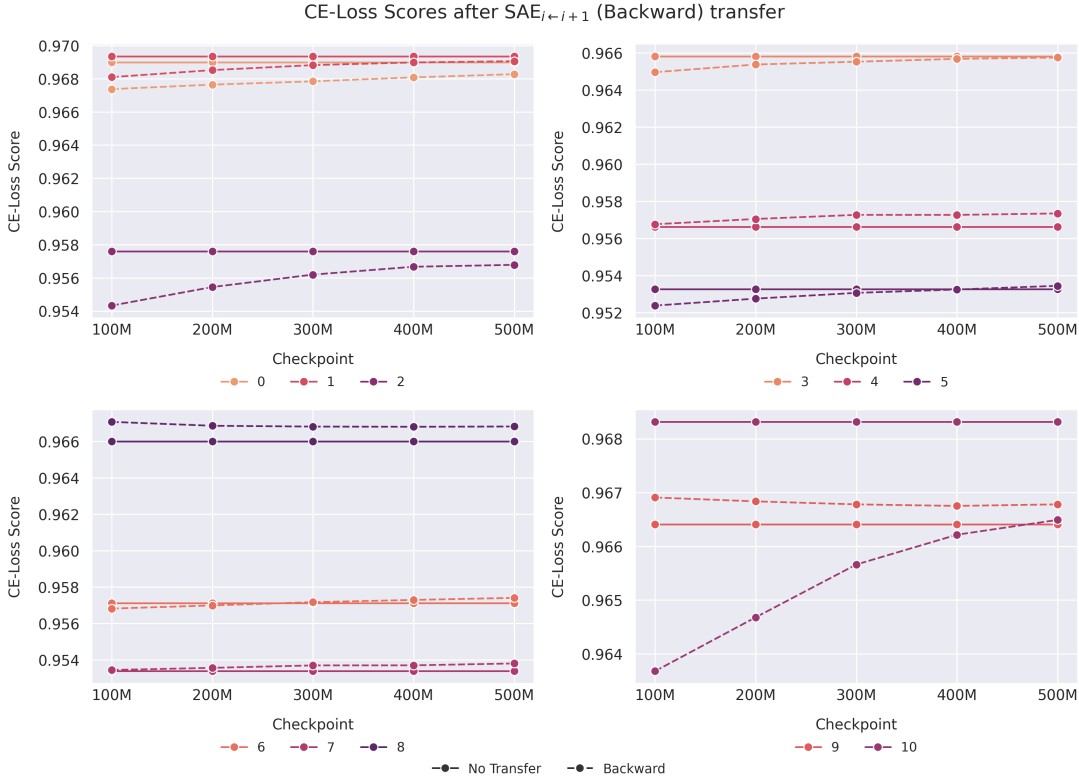

Figure 13: Detailed per-layer CE-Loss Score over time (Checkpoint) after Backward Transfer.

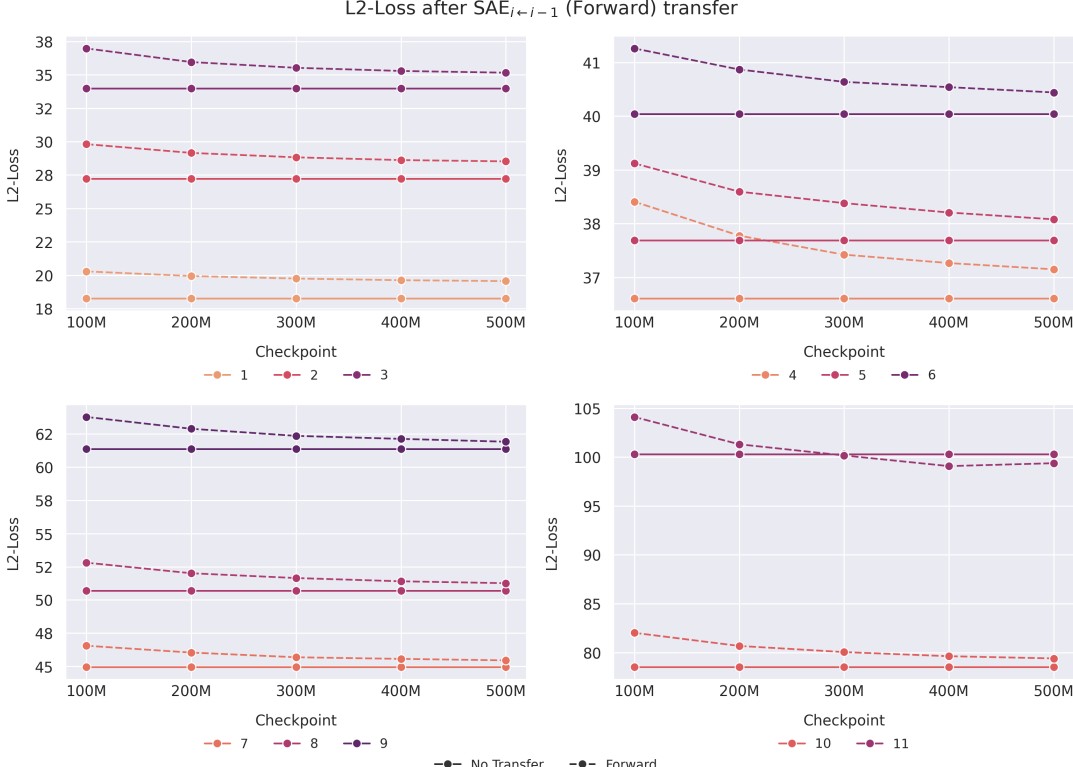

Figure 14: Detailed per-layer $L2$-Loss over time (Checkpoint) after Forward Transfer.

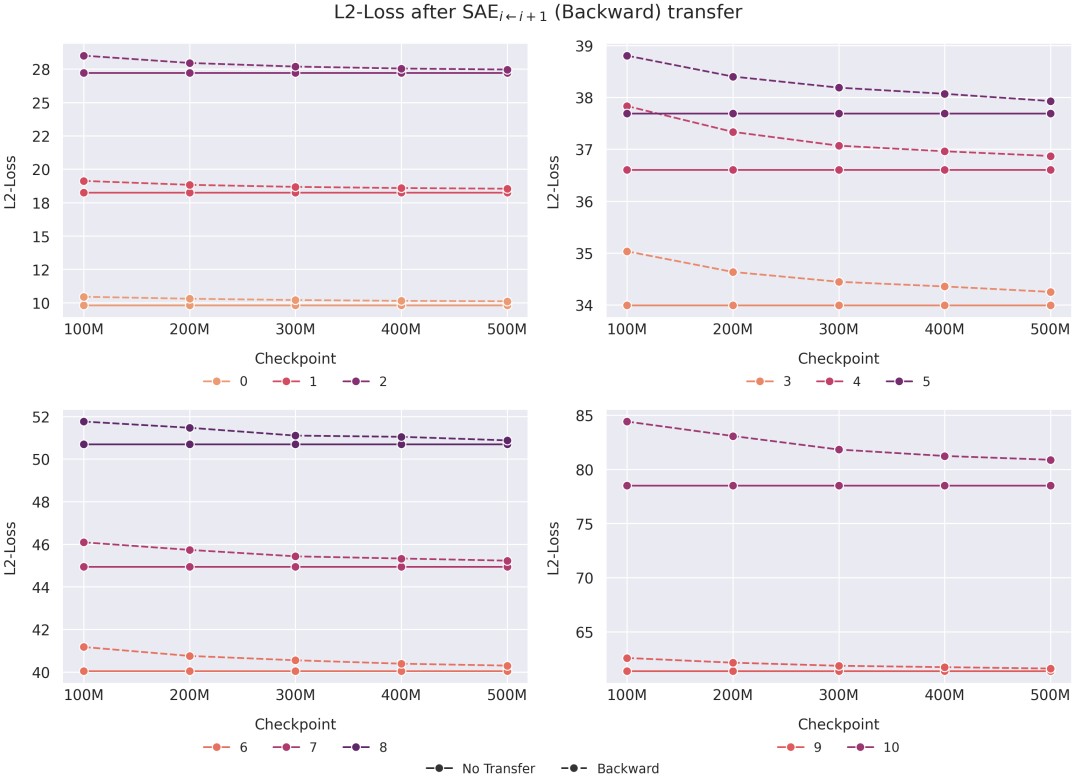

Figure 15: Detailed per-layer $L2$-Loss over time (Checkpoint) after Backward Transfer.

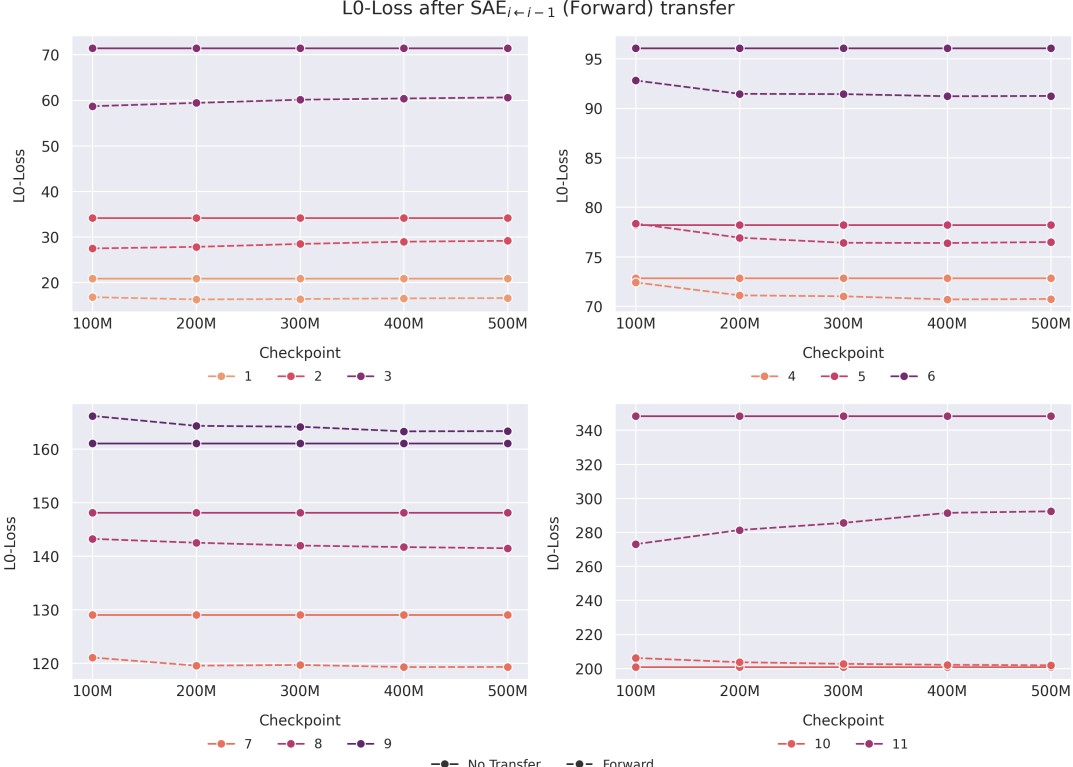

Figure 16: Detailed per-layer $L0$-Loss over time (Checkpoint) after Forward Transfer.

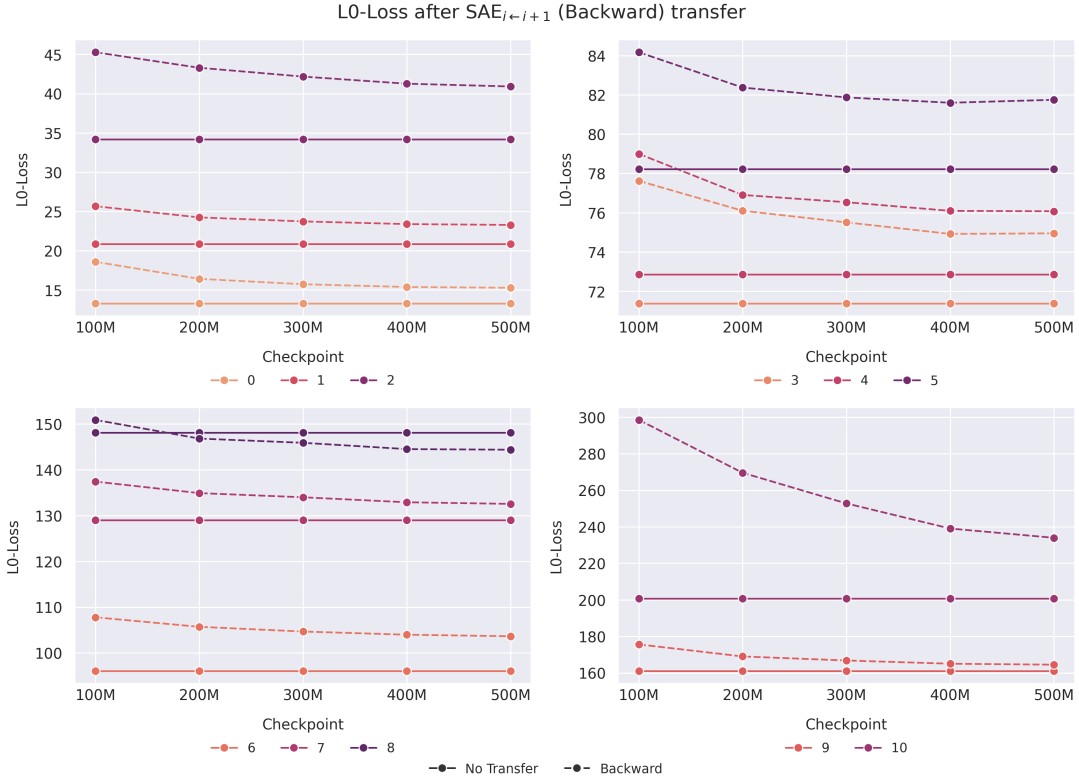

Figure 17: Detailed per-layer $L0$-Loss over time (Checkpoint) after Backward Transfer.

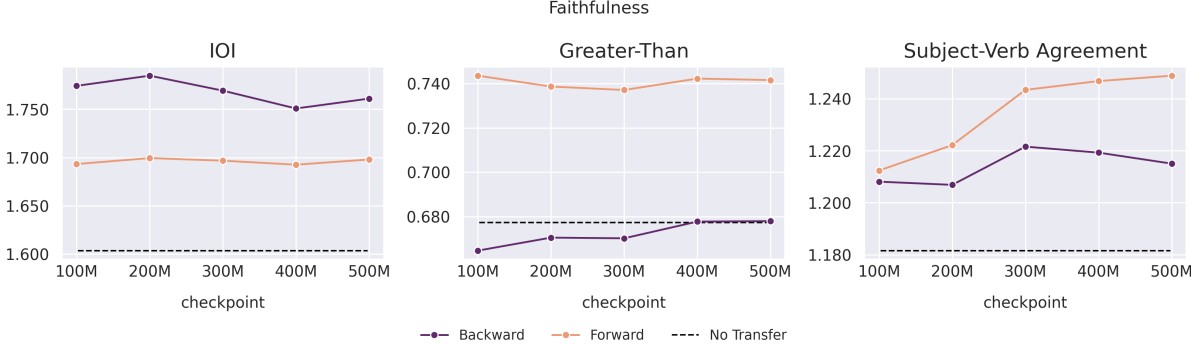

Figure 18: Faithfulness over time (Checkpoint) averaged by layer and $N$ for the three downstream tasks.

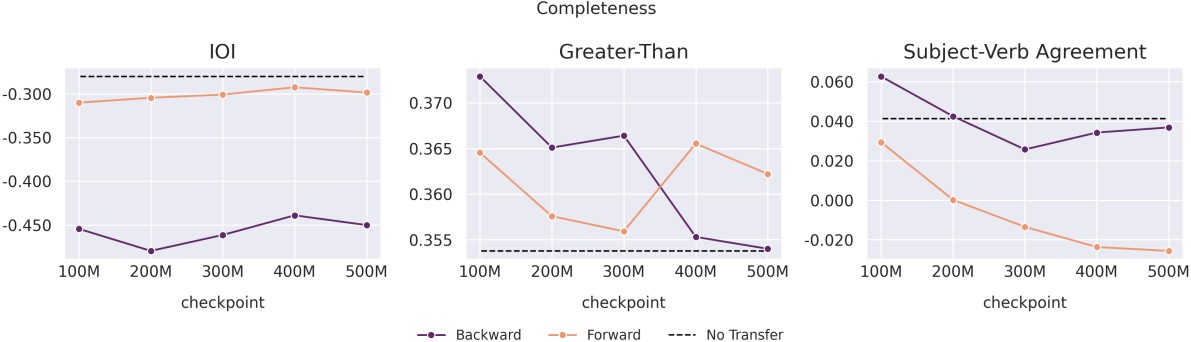

Figure 19: Completeness over time (Checkpoint) averaged by layer and $N$ for the three downstream tasks.

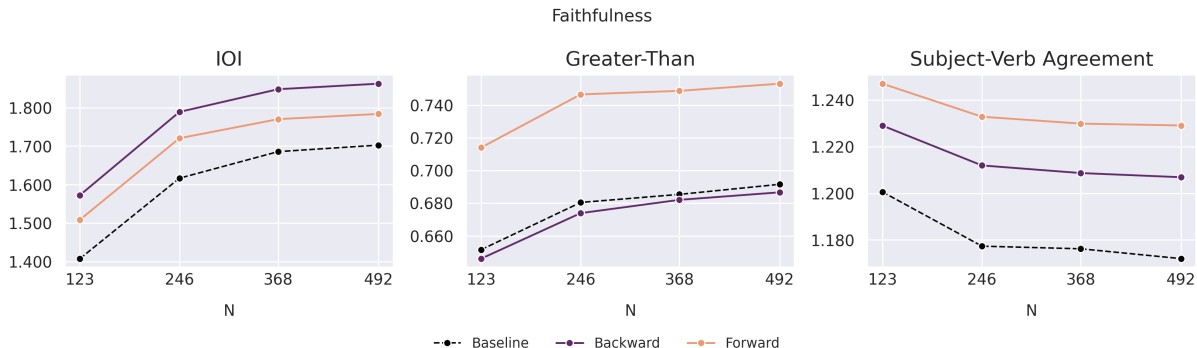

Figure 20: Faithfulness over $N$ averaged by layer and time (Checkpoints) for the three downstream tasks.

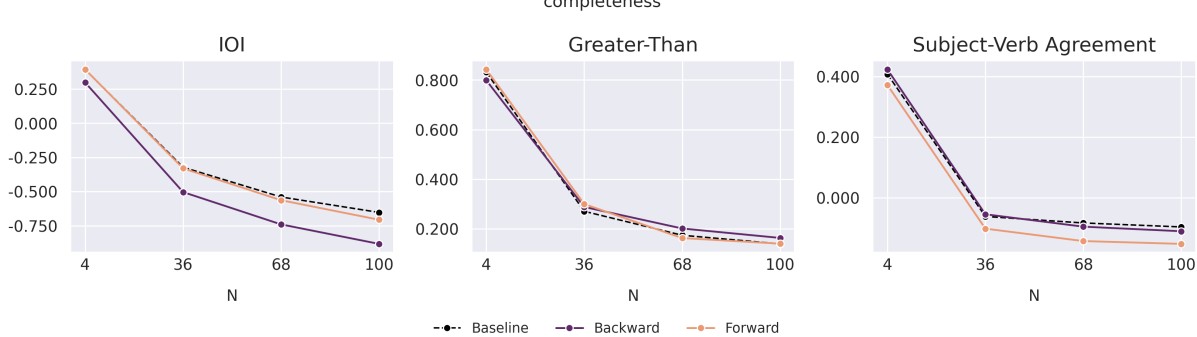

Figure 21: Completeness over $N$ averaged by layer and time (Checkpoints) for the three downstream tasks.

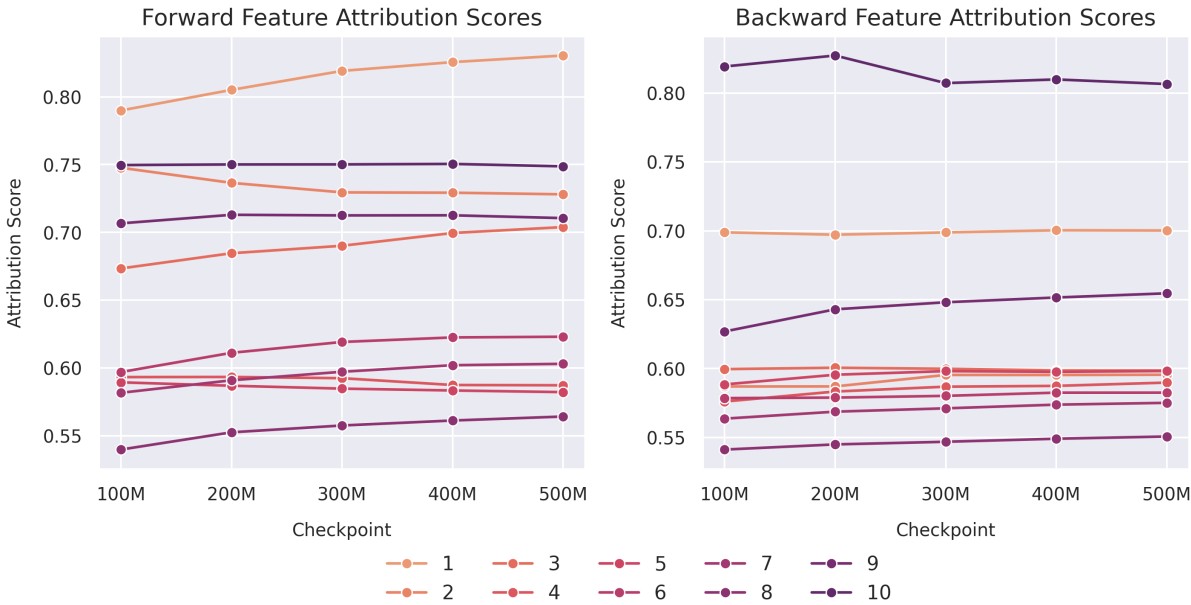

Figure 22: Detailed per-layer feature averaged Logits Attribution scores over time (Checkpoint), as defined in Equation 11.

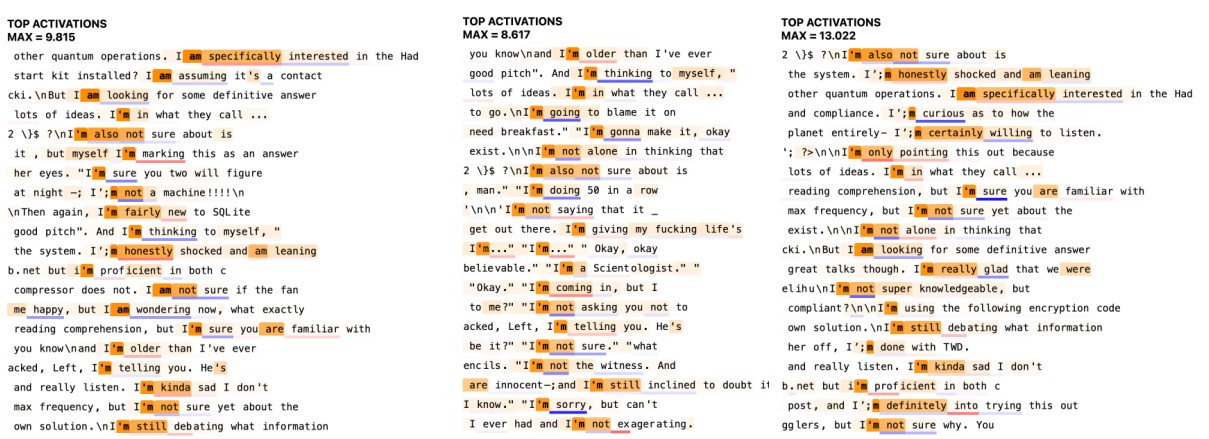

Figure 23: Comparison of top activations of feature 949 across layer 8 SAE and two transfer SAEs pre-trained on the former. $SAE_8$ (Left), $SAE_{7\leftarrow 8}$ (Middle), $SAE_{9\leftarrow 8}$ (Right). Evidence of feature transfer across three layers.