# OpenReview forum: "Accelerating Sparse Autoencoder Training via Layer-Wise Transfer Learning in Large Language Models"
_EMNLP/2024/Workshop/BlackBoxNLP — BlackboxNLP 2024_

### Official Review · Reviewer_JH3T · 2024-09-10

**Overall Assessment:** 5
**Confidence:** 3

**Best Paper:**

3

**Best Paper Justification:**

Makes solid contributions in SAE training

**Comments Questions Suggestions And Typos:**

See weakness

**Paper Summary:**

The authors conduct an extensive analysis of Sparse Autoencoders (SAE) training methodologies. They demonstrate that transfer learning across layers of an LLM helps in faster training and improves on metrics - a very useful insight. In addition exhaustive experiments on model checkpoints, forward-backward transferability and the interpretability itself makes this a strong submission.

**Summary Of Strengths:**

1. Exhaustive experiments with a variety of metrics considered.
2. Insightful observations on the transferability of SAE (forward-backward) makes a solid technical contribution.
3. Result presentation is good.

**Summary Of Weaknesses:**

1. The metrics can be better explained and summarized with a paragraph on their importance.
2. Some ablation studies on the dimension of the SAE might be helpful on different LLMs

---

### Official Review · Reviewer_YtyH · 2024-09-10

**Overall Assessment:** 4
**Confidence:** 4

**Best Paper:**

1

**Best Paper Justification:**

N/A

**Comments Questions Suggestions And Typos:**

L368: " has been then"
L476: "mcuh"

A more clear presentation of how much compute, in whatever metric makes sense (FLOPs, steps) would be helpful somewhere

**Paper Summary:**

This paper is motivated by the very high computational costs of training Sparse Autoencoders (SAE) from scratch, and proposes to alleviate this issue by reusing the weights from SAEs trained on adjacent layers. This is intuitive, and works very well in terms of reconstruction loss, mean max cosine similarity, and human evaluations. The authors show that finetuning the transferred weights to better fit the destination layer activations can even improve the SAEs beyond the quality of a from-scratch SAE in some settings. The authors are able to validate their hypothesis pretty convincingly, and provide a new way to speed up SAE training.

**Summary Of Strengths:**

* This paper presents novel and mostly thorough findings on the transferability of SAEs trained on different layers within the same network. While this is perhaps expected, this work provides the empirical backing, lays out exceptions to the transferability of layers like the last, and provides a recipe for applying this to new settings, which will greatly improve the computational cost of training SAEs.

* I believe the paper does a nice job recapitulating related work (in e.g,. inter-layer SAE transfer analysis) and situating their own findings in the literature. I think this paper will be useful to the community for this reason, especially since this topic is new, and SAE transfer is approached in a few different ways already

**Summary Of Weaknesses:**

* Because of the residual connection in these networks, it should be expected that some transfer holds, as vector additions are the only modifications made to every layer. I don't the experiments goes as deeply as they could for this reason. For example:
1.) What happens if you train one SAE from scratch (at say, layer L), transfer to L-1 and finetune it, transfer that to L-2, ..., and so on? Can the transfer bubble all the way down/up or do the error accumulate?
2.) What are some examples of features that require finetuning to learn? Are there any features that are /only/ learned by SAEs trained from scratch? This would be very useful to know for practitioners to actually want to use this method of training.
3.) re: residual connection, could you also transfer SAEs trained on MLPs of adjacent layers? This could be a useful lower bound, and I think *any* transfer from adjacent layers that isn't at non-adjacent layers in this setting is interesting.

* Some of the experimental results are presented very briefly. I'm not really sure what I'm supposed to take away from Figure 5. It seems confusing to me that the forward transfer has a higher DLA score or if this is necessarily a good thing. This might also be a little terse for readers less familiar with DLA, which comes mostly from circuit analysis work.

I think this represents solid work that will be useful for the community to engage with, though some deeper analyses may make it more exciting.

---

### Official Review · Reviewer_8bfs · 2024-09-10

**Overall Assessment:** 4
**Confidence:** 2

**Best Paper:**

1

**Best Paper Justification:**

-

**Comments Questions Suggestions And Typos:**

It would be interesting to run an experiment to consider where the transfer falls off. That is, instead of considering just the i and i+1 layers, would it be possible to extend to non-immediately adjacent layers and see where the results fail to hold?

**Paper Summary:**

This work aims to reduce the computational intensity of training Sparse Auto Encoders. They show that SAEs trained on one set of layers can serve as effective initialization for SAEs designed for closely related layers. This work also shows that SAEs exhibit partial transfer to adjacent layers.

**Summary Of Strengths:**

The paper is written with clarity and shows impressive results in the reduction of computational intensity. The perceived impact is high and could also be a starting point for a variety of new research ideas in this direction. It effectively demonstrates a transfer phenomenon and also leverages it practically to realize efficiency benefits.

**Summary Of Weaknesses:**

Is there an overemphasis on reconstruction scores? The main results (and perhaps relevant as well) are centered around reconstruction, would it make sense to also include details about performance metrics on downstream tasks?

The experiments are limited in scope with respect to model size, families and datasets. Since this is an empirical result that future practitioners need to know a priori if it will work for them, it might be important to see how well the findings hold up.

Most of my concerns are covered in the limitations section and could be good directions to explore future work.

---

### Decision · Program_Chairs · 2024-09-17

**Decision:**

Accept

**Comment:**

Reviewers agree that the paper's contributions are insightful, impactful, and well-presented. The topic is also clearly relevant to the theme of the workshop. There were some concerns regarding the presentation and choice of metrics, though evaluating SAEs (and mechanistic interpretability methods more broadly) is an established open problem. Regardless, it could be good to take some of these presentational suggestions into account when preparing the camera-ready.